# Field assessment in Cameroon of a reader of POC-CCA lateral flow strips for the quantification of *Schistosoma mansoni* circulating cathodic antigen in urine

Estelle Mezajou Mewamba[1], Arnol Auvaker Zebaze Tiofack[1], Cyrille Nguemnang Kamdem[1], Romuald Isaka Kamwa Ngassam[2], Mureille Carole Tchami Mbagnia[3], Oscar Nyangiri [4], Harry Noyes [5], Hilaire Marcaire Womeni[6], Flobert Njiokou[3,7], Gustave Simo [1]*

1 Molecular Parasitology and Entomology Unit, Department of Biochemistry, Faculty of Science, University of Dschang, Dschang, Cameroon, 2 Department of Biological sciences, Faculty of Science, University of Maroua, Maroua, Cameroon, 3 Parasitology and Ecology Laboratory, Department of Animal Biology and Physiology, Faculty of Science, University of Yaoundé I, Yaoundé, Cameroon, 4 College of Veterinary Medicine, Animal Resources and Biosecurity, Makerere University, Kampala, Uganda, 5 Centre for Genomic Research, University of Liverpool, Liverpool, United Kingdom, 6 Unité de Recherche de Biochimie, des plantes Médicinales, des Sciences alimentaires et Nutrition, University of Dschang, Dschang, Cameroon, 7 Centre for Research in Infectious Diseases, Yaoundé, Cameroon

* gsimoca@yahoo.fr, gustave.simo@univ-dschang.org

**Data Availability Statement:** All relevant data are within the manuscript.

## Abstract

### Background

Determining *Schistosoma mansoni* infection rate and intensity is challenging due to the low sensitivity of the Kato-Katz (KK) test that underestimates the true disease prevalence. Circulating cathodic antigen (CCA) excreted in urine is constantly produced by adult worms and has been used as the basis of a simple, non-invasive point of care test (POC-CCA) for *Schistosoma mansoni* infections. Although the abundance of CCA in urine is proportional to worm burden, the POC-CCA test is marketed as a qualitative test, making it difficult to investigate the wide range of infection intensities. This study was designed to compare the prevalence and intensity of *S. mansoni* by KK and POC-CCA and quantify, on fresh and frozen (<-20˚C) urine samples, CCA using the visual scores and the ESEquant LR3 reader.

### Methodology

Stool and urine samples were collected from 759 school-aged children. The prevalence and intensity of *S. mansoni* were determined using KK and POC-CCA. The degree of the positivity of POC-CCA was estimated by quantifying CCA on fresh and frozen urine samples using visual scores and strip reader. The prevalence, the infection intensity as well the relative amounts of CCA were compared.

### Results

The *S. mansoni* infection rates inferred from POC-CCA and KK were 40.7% and 9.4% respectively. Good correlations were observed between infection intensities recorded by; i)

**Funding:** This study constitutes output from the TrypanoGEN+ project; grant number H3A/18/004 funded by the African Academy of Sciences through the "Alliance for Accelerating Excellence in Science in Africa (AESA)" as part of the H3Africa consortium. EMM, AAZT, ON, HN, FN and GS were funded by H3A/18/004. CNK by MTN/OCEAC project; HMW by the University of Dschang; RIKN and MCTM respectively by the University of Maroua and the University of Yaounde 1. The funders played any role in the study design, data collection and analysis as well as in the decision to publish or in the preparation of the manuscript.

**Competing interests:** The authors have declared that no competing interests exist.

the reader and visual scoring system on fresh (Rho = 0.89) and frozen samples (Rho = 0.97), ii) the reader on fresh urine samples and KK (epg) (Rho = 0.44). Nevertheless, 238 POC-CCA positive children were negative for KK, and sixteen of them had high levels of CCA. The correlation between results from the reader on fresh and frozen samples was good (Rho = 0.85). On frozen samples, CCA was not detected in 55 samples that were positive in fresh urine samples.

## Conclusion

This study confirmed the low sensitivity of KK and the high capacity of POC-CCA to provide reliable data on the prevalence and intensity of *S. mansoni* infections. The lateral flow reader enabled accurate quantification of CCA under field conditions on fresh and frozen urine samples with less time and effort than KK.

### Author summary

Diagnosis of schistosomiasis has relied on the Kato-Katz technique which remains challenging due to its low sensitivity. To overcome this limitation, the Point-of-care-Circulating Cathodic Antigen (POC-CCA) test has been developed to detect CCA produced by adult living worm. However, this test is sold for qualitative use only because it is difficult to estimate the intensity of the positive band by eye. This study was designed with the aim of comparing the prevalence and intensity of *S. mansoni* infections by KK and POC-CCA and quantifying under field conditions on fresh and frozen (<-20˚C) urine samples, CCA using the visual scores and the ESEquant LR3 reader. We conducted the KK and POC-CCA tests on stool and urine samples collected from SAC (5 to 4 years) in Makenene, Cameroon. Our results showed discrepancies between results from KK and POC-CCA test. The numerical values generated by the reader made it possible to avoid subjective visual interpretation of POC-CCA results. This study also identified children with high levels of CCA in their urine but without schistosome eggs in their stools. The good correlation observed between results obtained on fresh and frozen urine samples confirmed that POC-CCA test can be used on samples stored for one year at -20˚ C.

## Introduction

Schistosomiasis is one of the most common neglected tropical diseases (NTD) and remains a major public health problem in many tropical and subtropical countries [1]. Approximately 700 million people are at risk of infection and more than 200 million are currently infected worldwide [2,3]. About 280,000 annual deaths are related to schistosomiasis and 3.3 million disability adjusted life years lost are due to this disease [4,5]. Beyond its public health importance, schistosome infections are regarded as one of the parasitic diseases with greatest socio-economic impact [6].

Although several strategies including preventive chemotherapy (PCT), water sanitation and hygiene (WASH), snail control, and the dissemination of information, education, and communication (IEC) have been developed to fight against schistosomiasis [7], the control of this disease relies mainly on PCT in which Praziquantel (PZQ) is distributed to school-aged children (SAC) once a year, twice a year or every 2 years depending on the current disease

prevalence in the endemic area [8,9]. For PCT, no diagnosis is made in each child before mass drug administration (MDA) of PZQ. However, the frequency of the MDA or the treatment rounds depends entirely on the disease prevalence in each community. Determining the accurate disease prevalence and the infection intensity in different communities with reliable tools appears therefore as the keystone for monitoring progress towards control of schistosomiasis in endemic areas.

The Kato Katz technique (KK) relies on the microscopic examination of faecal samples for the detection of schistosome eggs and has been routinely used to estimate the prevalence of schistosome infections in various communities [3,10,11]. It is recognised as being a direct, specific and low-cost technique that is relatively simple to perform under field conditions with limited health facilities [12,13,14,15,16,17]. In children of western Kenya for instance, Worrell et al. [17] have shown that US$ 6.89 are spent per-person for a single KK against US$7.26 for POC-CCA. Although these analyses took into account the costs linked to labor, transportation and supplies, they didn't consider other costs (susceptible to increase the KK cost) like those related to the training and retaining capable microscopists. Nevertheless, the KK test is known to miss many infections, especially in individuals harbouring low intensity of schistosome infections [18,19,20,21]. On the basis of KK data, the disease prevalence is generally underestimated, and this will have an impact on the implementation of PCT programmes. To overcome the low sensitivity of KK, immunochromatological diagnostic methods based on the detection of circulating cathodic (CCA) or anodic antigens (CAA) have been developed to detect actively secreted schistosome antigens [13,22,23,24,25]. The Point of Care Circulating Cathodic Antigen (POC-CCA) test was developed using a monoclonal antibody to detect *Schistosoma* CCA in urine [25]. CCA in the urine is evidence of the presence of adult worms even if *S. mansoni* eggs cannot be found in stool [26]. POC-CCA has been shown to be highly sensitive and specific for the detection of active *S. mansoni* infections [27]. It has therefore been recommended for the surveillance and mapping of intestinal schistosomiasis due to *S. mansoni* [23,25,28,29,30]. The amount of CCA released in the urine is directly related to the quantity of adult worms present in each individual [31,32,33]. In addition, POC-CCA is non-invasive, easy to handle and to perform under field conditions and requires minimal practical training for its implementation [12,13,34,35]. The intensity of the positive band on the POC-CCA strip is proportional to the concentration of CCA in the sample. However, the POC-CCA test was developed as a qualitative test since it is difficult to estimate the intensity of the positive band by eye and there can be significant variation in estimates of band intensity by different people [26]. To overcome these difficulties, this study was designed with the aim of comparing the prevalence and intensity of *S. mansoni* by KK and POC-CCA and accurately quantify, on fresh and frozen (<-20˚C) urine samples, CCA using the visual scores and the ESEquant LR3 reader.

## Methods

### Ethical statement

The study was approved by the National Ethics Committee for research on human health of the Ministry of Public Health of Cameroon with the reference number N˚2019/02/1144/CE/CNERSH/SP. The review board of the Molecular Parasitology and Entomology Subunit of the Department of Biochemistry of the Faculty of Science of the University of Dschang gave its approval. The field survey was conducted in schools with the approval of the administrative authorities, school inspectors, directors and teachers. Parents or guardians of participating children approved their participation by signing the informed consent form on their behalf. In addition, children of 10 to 14 years signed an assent form while for younger ones, only the

consent form signed by their parents or guardians was considered. After detailed explanation of the objectives, the procedures and the potential risks and benefits, each child was free to choose whether to participate in the study. Data were anonymised during analyses.

Results of parasitological and immunochromatological tests were communicated to parents or guardians and all children found with schistosome infections were treated with a single dose of PZQ (40mg/Kg body weight) following WHO recommendations [16].

## Study area

This study was carried out at Makenene which is a rural locality of the Mbam and Inoudou Division of the Centre Region of Cameroon. Makenene is located about 200 km to the north-west of Yaoundé and has about 16,000 inhabitants. It belongs to the Bafia Health District and is delimited in the north by the council of Massagam, the south by the council of Yabassi and Ndikimineki, the east by the council of Deuk and Konyambetta and finally in the west by the council of Tonga. It has an equatorial climate with 4 seasons: two dry seasons from November to March and from mid-May to mid-August and two rainy seasons from August to November and March to July. Its dense hydrographic network contains several rivers including the Mock, Makombé, Makongo, Managa, Mefom, Niep, Bokokeut, Kyakan, Mayi, Molo, Makam, Sinsam, Bambi, Djanka and Noum. Inhabitants of this locality practice petty trading and farming and the main crops are cassava, corn, groundnuts, yam, cocoa and palm nuts. *Schistosoma mansoni* infections have been reported in villages of Makenene for more than 30 years with prevalence ranging from 82% in 1987 to 18% in 2001 [36,37,38]. Mass administration of PZQ is yearly performed in this locality.

## Study design and population

This cross-sectional study was conducted in October 2019. Participants were children attending the Bilingual Public School of Baloua (EPB,), the Public School of Makenene (EPM) and the Public School of Ngokop (EPN). These schools were randomly selected and in each of them, stool and urine samples were collected from all children who accepted to participate in the study. Each child who participated to the study was between 5 and 14 years old. The sample size (n) was calculated using the following formula: $n = Z^2 P (1 - P)/d^2$, Where n = estimated the sample size, Z is critical value (1.96) at 95% confidence level, P is the prevalence of intestinal schistosomiasis in Makenene (69.8% inferred by POC-CCA test [28]) and d is precision or margin of error (5%). With this formula, the estimated sample size was computed to be 324.

## Sample collection

The directors and teachers at each school were given an orientation before sampling began and the objectives of the study were explained to them. Children were informed about the sampling day by their teachers. Each child who provided assent and had informed consent from their parents or guardians was invited to provide urine and stool samples in clean and well-labelled plastic containers that were given the day of the sampling. Approximately 50 ml of urine and 5 to 7g of stool were collected from children between 9:00 am and 2:00 pm. These samples were immediately transferred to the local health centre where antigen detection using POC-CCA and KK tests were performed on urine and stool samples respectively. The samples were kept at room temperature and completely processed within 12 hours following their collection.

## Detection of schistosome infections using the parasitological method

Schistosome eggs were detected using the protocol described by Katz et al. [10]. From each stool sample, a single KK thick smears slide was prepared with 41.7 mg of stool using the Sterlitech kit (Lot: XGACAI). Briefly, a small amount of stool was transferred onto a piece of scrap paper. Thereafter, a nylon mesh was pressed on the top of the faecal sample and a small plastic spatula was used to scrap the sieved material of the nylon screen. Subsequently, the well of the KK template was placed on a clean microscopic slide and completely filled with sieved faecal material. The template was removed without disturbing the calibrated faecal material. Thereafter, the slide was covered with a cellophane strip soaked in glycerol-malachite green solution. The stool was spread onto a thick smear by inverting the microscope slide and pressing the stool sample against the cellophane on a smooth surface. Slides were searched for *S. mansoni* eggs after 24 hours and the infection intensity was calculated and expressed as the number of eggs per gram of stool (EPG). Samples were classified as having light, moderate or heavy infections when the number of eggs per gram of stool was respectively <100 EPG, 100–400 EPG and >400 EPG [16].

## Detection of circulating cathodic antigens of schistosomes

**Detection of CCA in fresh urine samples.**   Circulating cathodic antigens were detected using POC-CCA (Rapid Medical Diagnostics, Pretoria, South Africa, batch no 190411032) which is a rapid diagnostic test for *S. mansoni* infections. The POC-CCA test was performed according to the manufacturer instructions. Briefly, two drops of urine sample were put in the well of the POC-CCA cassette. After complete absorption of the urine sample for exactly 20 minutes, the POC-CCA cassette was visualized. A test was considered valid if the control line turned a dark pink colour. If not, the sample was re-tested with a new cassette. Any cassette that was not read at 20 minutes was considered as invalid and therefore, re-tested [39].

**Quantification of CCA on POC-CCA cassettes.**   To estimate the amount of CCA on the POC-CCA cassette and therefore in the urine sample of each child, two approaches were used: 1) the ESEquant reader (Fig 1) that quantifies the amount of CCA on each cassette and 2) visual scoring by reference to standard samples provided by Department of Parasitology, Leiden University Medical Center, Leiden, The Netherlands. The first approach used the ESE-Quant LR3 reader (Qiagen, Germany) (Fig 1) that is designed to quantify target analytes on Lateral Flow Test strips including the CCA strips. During the reading process, each cassette was introduced into the drawer (Fig 1) of the ESEQuant LR3 reader (Qiagen, Ref: ESLR11-MB-6401). The intensity of CCA on each POC-CCA cassette was estimated and expressed in millivolts following the manufacturer's instructions [40]. Each measurement expressed in millivolts was proportional to the intensity of the band reflecting the level of CCA on the strip or in the urine sample. Children having high CCA band intensity were considered as having high level of CCA and therefore, high worm burden.

For the visual scoring system, each POC-CCA cassette was removed from the drawer (Fig 1; Qiagen, Reference number ESLR11-MA-1140) of the ESEquant LR3 reader and the intensity of CCA was scored by another person using G-scores as described by Casacurberta et al. [41]. In this scoring system, the intensities of bands appearing on POC-CCA cassette were compared to those of G-scores which are series of 10 artificial cassettes containing inkjet-printed strips with different line intensities named G1 to G10 (Table 1). The quantification of CCA on POC-CCA cassettes using G-scores is an innovative and user-friendly scoring system that records the intensity of CCA as 0, "trace", 1+, 2+ or 3+ depending of their correspondence with G-scores (Table 1).

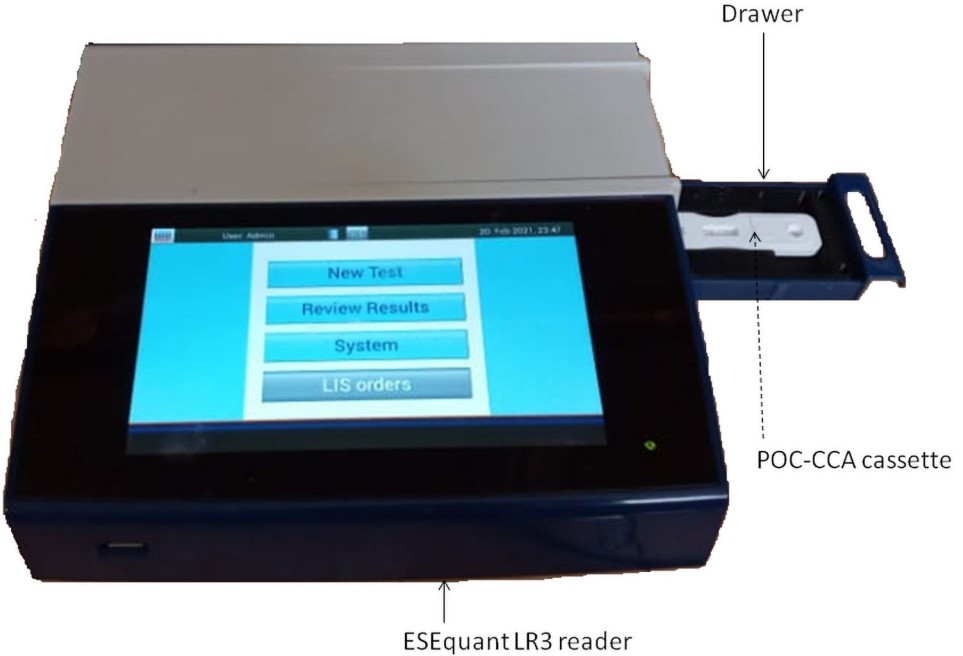

**Fig 1. ESEquant LR3 lateral flow strip reader.**

**Detection of CCA on stored urine samples.**   Some urine samples that where positive for POC-CCA test were kept at 4˚C in the field for up to four days. In the laboratory, these samples were stored at -20˚C for one year. To quantify CCA on stored samples, each urine sample was removed from the freezer and allowed to thaw at room temperature for at least one hour. Thereafter, CCA was detected on each thawed sample using POC-CCA test as described above.

## Statistical analysis

Data were computed using XLSTAT 2020.2.1 and R 3.6.1 software. The chi-square and Fisher exact tests were used to compare the *S. mansoni* infection rates between age and sex. The infection intensity between sexes was compared using the Student's *t*-test. Linear regression was used to check for any association between infection intensity, age and sex. The test was

**Table 1.  G scores and their corresponding visual score [41].**

| G-score | Visual score |
| --- | --- |
| **G1** | 0 |
| **G2** | Trace |
| **G3** | Trace |
| **G4** | 1+ |
| **G5** | 1+ |
| **G6** | 2+ |
| **G7** | 2+ |
| **G8** | 3+ |
| **G9** | 3+ |
| **G10** | 3+ |

considered significant for a P value below 0.05. The Spearman correlation test was used to determine the relationship between the amount of CCA estimated by the reader and the G-scores on fresh and frozen urine samples as well as the infection intensity expressed as EPG. The Pearson correlation test was used to determine the correlation coefficients between results for frozen and fresh urine samples for the data generated by the reader and the visual scoring system. The kappa values with 95% CIs were used to determine the concordance between KK and POC-CCA. These values were interpreted according to the classification of [42].

## Results

### Field survey

A total of 759 children aged 5 to 14 years were examined: 385 (50.7%) males and 374 (49.3%) females. Amongst these children, 387 were from the Bilingual primary school of Baloua, 208 from the public school of Makenene and 164 from the public school of Ngokop. The mean age of children examined was 8.55±2.32 (95% CI 8.34–8.76) years: 8.83±2.36 (95% CI 8.53–9.12) for males and 8.26±2.23 (95% CI 7.97–8.56) for females. Boys were significantly older than girls ($p < 0.001$, $t = 3.38$).

### Infection rate of *S. mansoni*

From the 759 children who were examined, 71 (9.4%) (95% CI 7.3–11.79) and 309 (40.7%) (95% CI 36.3–45.51) were respectively positive by the KK and POC-CCA tests.

Forty-two (10.9%) boys were positive for KK against 29 (7.8%) girls (Table 2). The highest number of positive KK (20%) was found in children of 14 years and the lowest (3.38%) in those of 6 years (Table 3). Comparing the *S. mansoni* infection rates revealed by KK, no significant difference ($P = 0.14$, $\chi^2 = 2.23$, $df = 1$) was observed between girls and boys (Table 2) or between ages ($P = 0.06$, $\chi^2 = 16.37$, $df = 9$) (Table 3).

One hundred and fifty-six (50.5%) boys were positive for POC-CCA against 153 (49.5%) girls (Table 2). The highest number of positive POC-CCA (49.0%) was found in children of 9 years and the lowest (30.2%) in those of 5 years (Table 3). Comparing the *S. mansoni* infection rates revealed by POC-CCA, no significant difference ($P = 0.91$; $\chi^2 = 0.01$, $df = 1$) was observed between boys and girls (Table 2) and between ages ($P = 0.64$, $\chi^2 = 6.98$, $df = 9$) (Table 3).

### Intensity of *S. mansoni* infections in stool and fresh urine samples

Of the 71 children with *S. mansoni* eggs by KK, 28 (39.4%), 23 (32.4%) and 20 (28.2%) were found respectively with light, moderate and high infection intensity. The mean EPG was

**Table 2.** Prevalence and intensity of *S. mansoni* infections by KK and POC-CCA in boys and girls.

|  | N | POC-CCA (%) | 95% CI | Mean CCA value (mv) | 95% CI | KK (%) | 95% CI | Mean EPG | 95% CI |
|---|---|---|---|---|---|---|---|---|---|
| **Male** | 385 | 156 (40.5) | 34.41–47.4 | 188.77±158.64 | 186.62–190.94 | 42 (10.9) | 7.86–14.75 | 564±1390.8 | 556.84–571.23 |
| **Female** | 374 | 153 (40.9) | 34.68–47.93 | 153.74 ± 129.08 | 151.79–155.72 | 29 (7.7) | 5.19–11.136 | 512.28±867.93 | 504.07–520.58 |
| **P-value** |  | 0.91[a] |  | 0.03[b] |  | 0.14[a] |  | 0.86[b] |  |
| **$\chi^2$** |  | 0.01 |  | - |  | 2.23 |  | - |  |
| **t** |  | - |  | 2.13 |  | - |  | **0.18** |  |

N: Number of children analysed; POC-CCA: Number of children positive for POC-CCA test; KK: Number of children positive for the Kato Katz test; CI: Confidence interval, EPG: egg per gram of stool;

[a]: P-value for chi-square test;

[b]: P- value for Student's *t*-test.

**Table 3. Prevalence and intensity of *S. mansoni* infections by KK and POC-CCA according to age.**

| Age | N | POC-CCA (%) | 95% CI | Mean POC-CCA values (mV) | 95%CI | KK(%) | 95% CI | Mean EPG | 95% CI |
|---|---|---|---|---|---|---|---|---|---|
| 5 | 43 | 13(30.2) | 16.1–51.7 | 112.98±121.06 | 107.22–118.85 | 2 (4.65) | 0.56–16.80 | 24±0.00 | 17.696–31.82 |
| 6 | 148 | 60(40.5) | 30.94–52.18 | 117.48±104.33 | 114.74–120.24 | 5 (3.38) | 1.09–7.88 | 648±707.50 | 625.88–670.7 |
| 7 | 121 | 49(40.5) | 29.96–53.54 | 181.17±152.85 | 177.41–184.97 | 9(7.44) | 3.40–14.12 | 290.67±281.80 | 279.63–302.02 |
| 8 | 73 | 28(38.4) | 25.49–55.44 | 128.77±123.85 | 124.58–133.02 | 6 (8.22) | 3.01–17.89 | 384±660.41 | 368.48–400.01 |
| 9 | 100 | 49(49) | 36.25–64.78 | 187.52±164.49 | 183.7–191.38 | 12 (12) | 6.2–20.96 | 176±149.82 | 168.57–183.67 |
| 10 | 111 | 41(36.9) | 26.51–50.11 | 171.23±128.09 | 167.24–175.27 | 12 (10.81) | 5.60–18.88 | 1120±2568.95 | 1101.1–1139.1 |
| 11 | 64 | 24(37.5) | 24.03–55.8 | 235.03±161.05 | 228.91–241.21 | 9(14.06) | 6.43–26.69 | 728±1199.70 | 710.48–745.84 |
| 12 | 57 | 26(45.6) | 29.8–66.84 | 209.62±147.90 | 204.09–215.26 | 9(15.79) | 7.22–29.97 | 248±244.75 | 237.82–258.51 |
| 13 | 32 | 15(46.9) | 26.24–77.31 | 244.40±175.34 | 236.55–252.44 | 5(15.63) | 5.07–36.46 | 849.6±562.24 | 824.24–875.54 |
| 14 | 10 | 4(40) | 10.9–102.4 | 251.53±198.13 | 236.2–267.53 | 2(20) | 2.42–72.25 | 876±899.44 | 835.46–918 |
| Total | 759 | 309(40.7) | 36.3–45.51 | 171.43±145.59 | 169.97–172.9 | 71(9.4) | 7.3–11.79 | 542.87±1197.86 | 537.47–548.32 |
| P | | 0.64[a] | | 0.00022[b] | | 0.06[a] | | 0.09[b] | |
| $\chi^2$ | | 6.98 | | - | | 16.37 | | - | |
| F | | - | | 40.36 | | - | | 3.53 | |

N: Number of children analysed; POC-CCA: Number of children positive for POC-CCA test; KK: Number of children positive for the Kato Katz test; CI: Confidence interval; EPG: egg per gram of stool;

[a]: P-value for chi-square test;

[b]: P- value for linear regression model.

564±1390.8 EPG (95% CI 556.84–571.23) in boys and 512.28±867.93 EPG (95% CI 504.07–520.58) in girls (Table 2). Comparing the mean EPG, no significant difference (*P = 0.86; t = 0.18*) (Table 2) was observed between boys and girls or between age groups (*P = 0.09; F = 3.53*).

For the detection of CCA of *S. mansoni* on fresh urine samples, results of visual scoring show that out of 309 children who were positive to POC-CCA, 172 (55.66%), 60 (19.42%), 37 (11.97%) and 40 (12.94%) had infection intensities classified respectively as "trace", 1+, 2 + and 3+. With the reader, the infection intensities varied from 40.03 mV to 633.5mV. The mean value of POC-CCA was 188.77±158.64 mV in boys and 153.74 ± 129.08 mV in girls (Table 2). Comparing the infection intensity according to sex using the Student's *t*-test, significant difference (*P = 0.03, t = 2.13*) was observed between boys and girls (Table 2). Using the simple linear regression test to compare infection intensity according to age, significant difference (*P = 0.0002; F = 40.36*) was also observed (Table 3). A good (adjusted $r^2$ = 0.81) correlation was observed between the infection intensity and age. The infection intensity seems therefore to increase with age (Fig 2). Although both sex and age were significantly associated with infection intensity when tested separately, when both age and sex were used as explanatory variables in a linear regression model, only age remained significantly associated with infection intensity (*P <0.0001, F = 17.45*) but sex was not (*p = 0.08, F = 3.10*).

## Comparison between Kato-Katz and POC-CCA results

From the 759 fresh urine and stool samples examined for CCA and *S. mansoni* eggs, concordant results between KK and POC-CCA were observed for 521 (68.6%) children in whom 71 (9.4%) and 450 (59.3%) were respectively positive and negative for both tests. Discordant results were obtained in 238 (31.4%) children who were all positive for POC-CCA, but negative by the KK test. All children positive by the KK test were also positive for POC-CCA. The

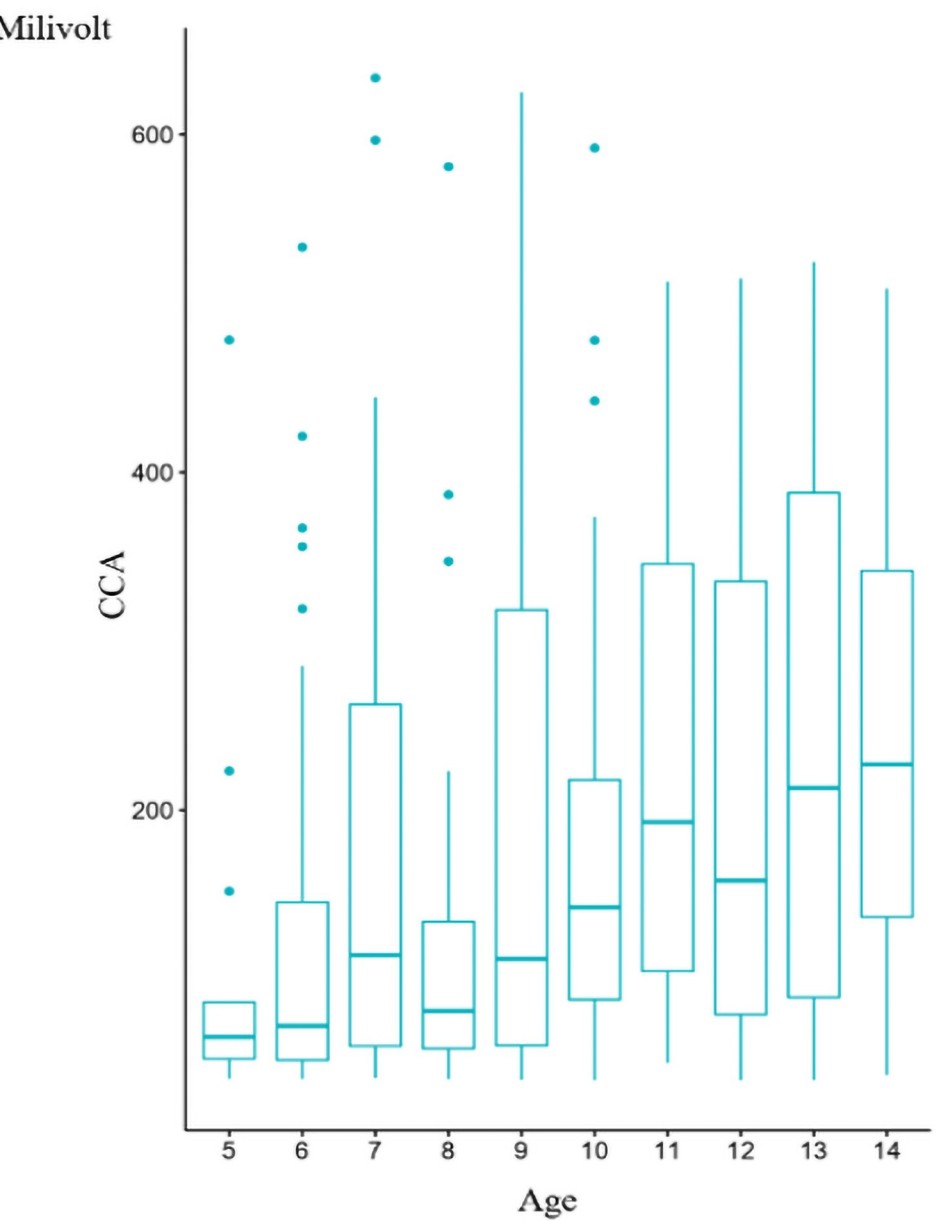

**Fig 2. Correlation between mean POC-CCA values on fresh urine samples and age of children.**

concordance index expressed as the Kappa coefficient was 0.26 (*P < 0.0001*; 95% CI: 0.21–0.31): indicating a fair strength of agreement between KK and POC-CCA results.

There was a positive correlation (rho Spearman's correlation coefficient = 0.44; *P < 0.0001*) between infection intensities of *S. mansoni* recorded in EPG (KK) and in millivolts (POC-CCA reader) for CCA on fresh urine samples. Most children with fresh urine samples showing high CCA values also had high number of eggs in their stools (Fig 3). However, 238 children who tested positive for the presence CCA in their fresh urine samples had no eggs in their stool samples (Fig 3). Amongst these 238 children, 16 (6.72%) had a high level of CCA of *S. mansoni* with CCA values from the reader ranging from 384.02mv to 633.5 mv and visually scored as

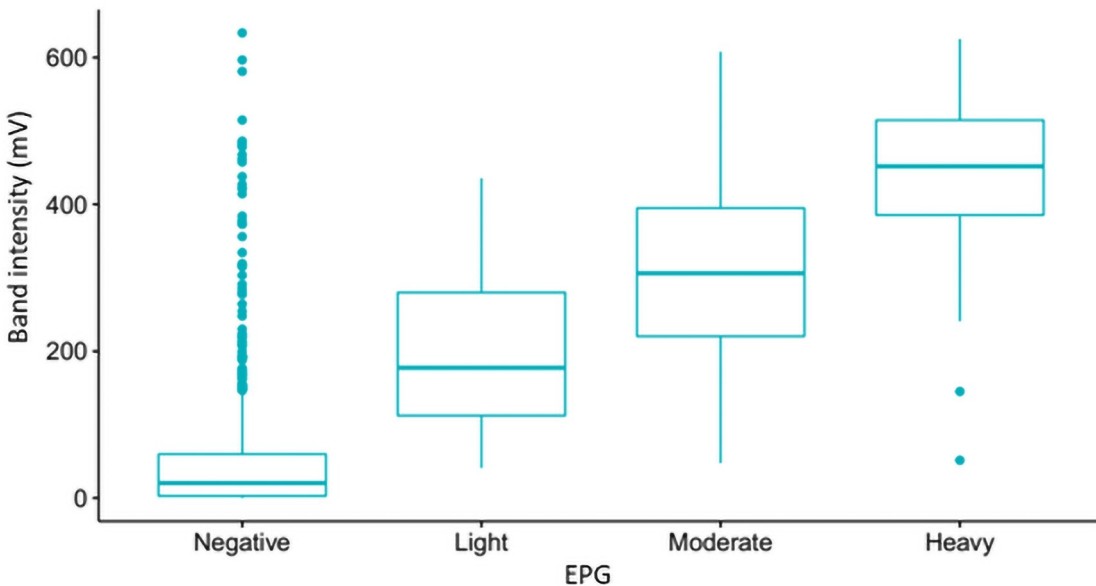

**Fig 3. Boxplot representing the relation between values of infection intensities from the ESEquant reader in millivolts and KK in eggs per grams of faeces (EPG).** Negative = 0 EPG; light <100 EPG; moderate = 100–400 EPG; heavy = >400 EPG.

3+. There is relatively large number of high POC-CCA band intensisties in some samples with negative KK tests but not in other groups (Fig 3).

Comparing the infection intensities resulting from the POC-CCA reader and the visual scoring, a strong and significant correlation (Spearman coefficient = 0.89; p < 0.0001) was observed. The POC-CCA reader revealed high amount of CCA in the strips of children for whom the visual scoring system scored at 3+ (Fig 4).

## Comparison of POC-CCA results generated from frozen and fresh urine samples

Of the 309 fresh urine samples that were positive for POC-CCA test, 215 of them were stored at -20°C. These samples were all retested after being stored for one year. Amongst these 215

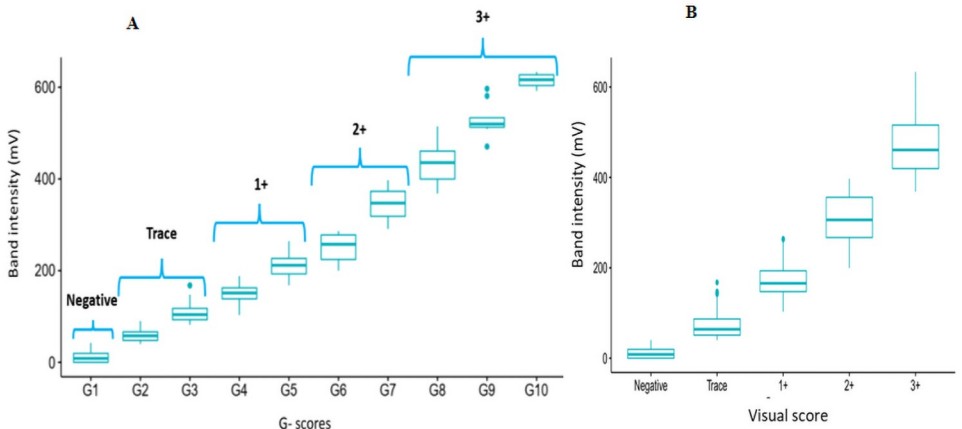

**Fig 4. Correlation between infection intensity from the POC-CCA tests using the ESEquant reader and the G-scores (Fig 4A) or their corresponding visual score (Fig 4B).**

**Table 4. Results of POC-CCA generated by visual scoring system on frozen and fresh urine samples.**

| Sample | Negative | POC-CCA Scores | | | |
|--------|----------|-------|----|----|----|
| | | Trace | 1+ | 2+ | 3+ |
| Fresh | 0 | 120 | 45 | 24 | 26 |
| Frozen | 55 | 68 | 21 | 45 | 26 |

samples and on the basis of results generated from fresh samples, 120 (55.81%), 45 (20.93%), 24 (11.16%) and 26 (12.09%) of them had scores classified respectively as Trace, 1+. 2+ and 3+ (Table 4). Results of POC-CCA tests on the 215 frozen urine samples are summarized in Table 4. Some variations were observed on the visual scores generated on fresh and frozen urine samples. From the 120 fresh urine samples classified as trace for instance, 49, 61 and 10 were scored on frozen samples as negative, trace and 1+ respectively. Out of 45 fresh urine samples classified as 1+, 4, 7, 10, 23 and 1 were scored on frozen samples respectively as negative, trace, 1+, 2+ and 3+. For the 24 fresh urines samples classified as 2+, 1, 1, 15 and 7 were scored on frozen samples respectively as negative, 1+, 2+ and 3+. Concerning the 26 fresh urines samples classified as 3+, 1, 7, and 18 were scored on frozen urine samples respectively as negative, 2+ and 3+. The number of urine samples scored as 2+ increased from 24 in fresh to 45 in frozen urine samples (Table 4).

Out of 55 negative frozen samples, 49 (89.1%) were scored as trace (fresh samples). For the 6 remaining, 4 (7.3%) were scored 1+, one (1.8%) 2+ and the last (1.8%) 3+. Comparing results generated on fresh and frozen urine samples, good and significant correlations were observed with the reader (Pearson rho = 0.85, $P < 0.0001$) (Fig 5) as well as the visual scoring system (Pearson rho = 0.84, $P < 0.0001$). The means of CCA on POC-CCA strips decrease when these samples were stored for one year at -20˚ C ($P < 0.001$, F = 6.00). Between fresh and frozen urine samples for instance, the mean values of POC-CCA decrease from: i) 69.32mv to 54.89 mV for samples with "trace" score; ii) 173.23mV to 150.92mV for those scored 1+; iii) 307.93mV to 275.25mV for samples with 2+ and; iv) 471.73mV to 398.67mV for samples having 3+.

Comparing the infection intensities generated by the reader and the visual scoring system on frozen urine samples, a strong and significant correlation (Spearman coefficient = 0.97; p < 0.0001) was observed (Fig 6). These results are in line with those obtained on fresh urine samples.

## Discussion

The *S. mansoni* infection rate of 40.7% inferred by POC-CCA is lower than 69.8% obtained 8 years ago in this same locality [28]. Regarding the infection rates by KK, our value of 9.4% is lower than 36.26% and 49% reported 7 and 25 years ago [37,43]. This reduction is in agreement with what is observed in most schistosomiasis endemic regions and could be explained by the the mass administration of PZQ initiated in schistosomiasis endemic areas of Cameroon since 2007 [43]. Indeed, following annual mass distribution of PZQ to SAC, a reduction of schistosomiasis prevalence from 81.60% to 41% has been reported from 1985 to 2010 [43].

The prevalence of *S. mansoni* inferred by POC-CCA (40.7%) is 4.3 times higher than the prevalence by KK (9.4%). These results confirm the low sensitivity of the KK test for the diagnosis of intestinal schistosomiasis in low disease prevalence and intensity settings [22,23,29,44]. In such settings, the daily fluctuations in egg excretion and the heterogeneous distribution of eggs within the stool sample could lead to misdiagnosis of some infected individuals [15,45]. For the present study, it is important to point out that the infection rate

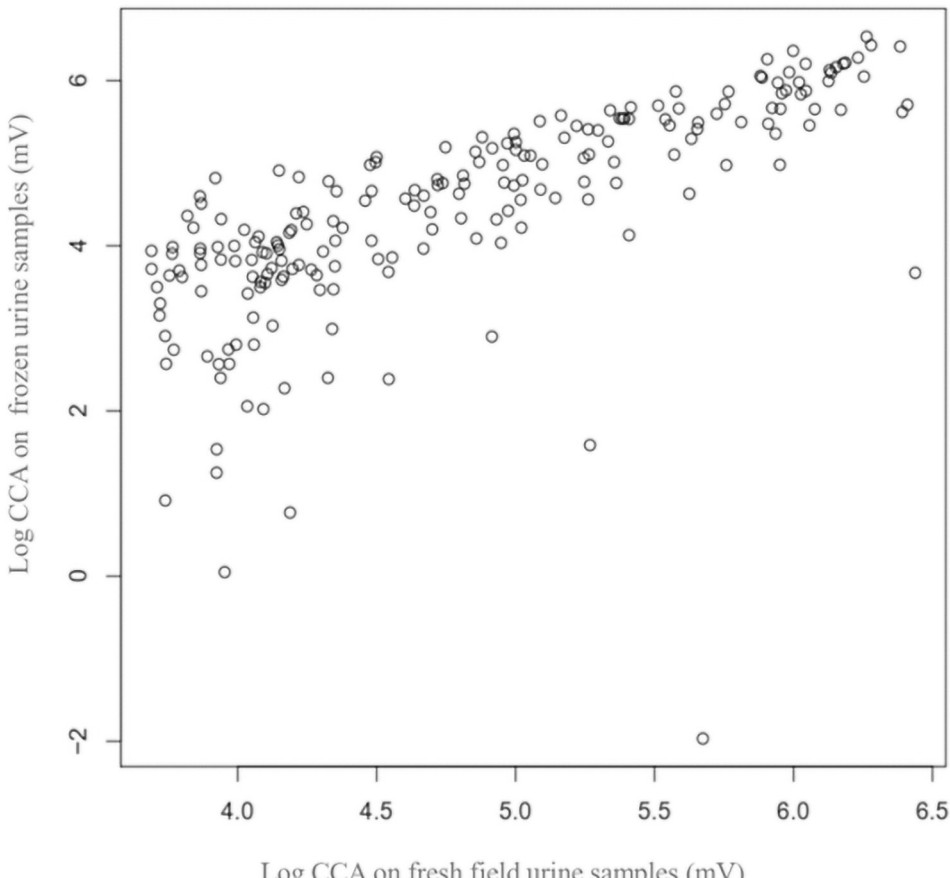

**Fig 5. Correlation between infection intensity generated by the reader on frozen and fresh urine samples.**

revealed by KK has been probably underestimated because only one KK slide was examined for each child. In such context, some children and especially those with low infection intensity of *S. mansoni* could pass undetected when only one KK slide is screened. This has been already demonstrated in previously studies [46].

Our results showing no significant difference in the *S. mansoni* infection rates according to sex are in agreement with those reported in some African countries such as Mali, Ethiopia, Kenya and the Democratic Republic of Confo [47,48,49,50,51]. But they contradict reports from other endemic countries such as Ghana [51], Central Sudan and Yemen where boys have been reported to be more infected than girls [52,53,54,55,56,57,58,59]. The discrepancy between results of different studies could be explained by some differences in the behaviour of boys and girls in different epidemiological settings. Although information regarding the activities performed by boys and girls has not been collected during the present study, it is likely that boys and girls are involved in the same risky activities since they swim, wash clothes and fetch water in the same places. They are consequently subjected to similar exposure to schistosomiasis infections.

Results of the present study highlighting no significant difference in the infection rates according to age are in line with those of Angora et al. [60] but differ from other studies where an increase in prevalence was observed with age. For instance, Sady et al. [54] reported higher *S. mansoni* prevalence in children over 10 years old than younger children. The discrepancy

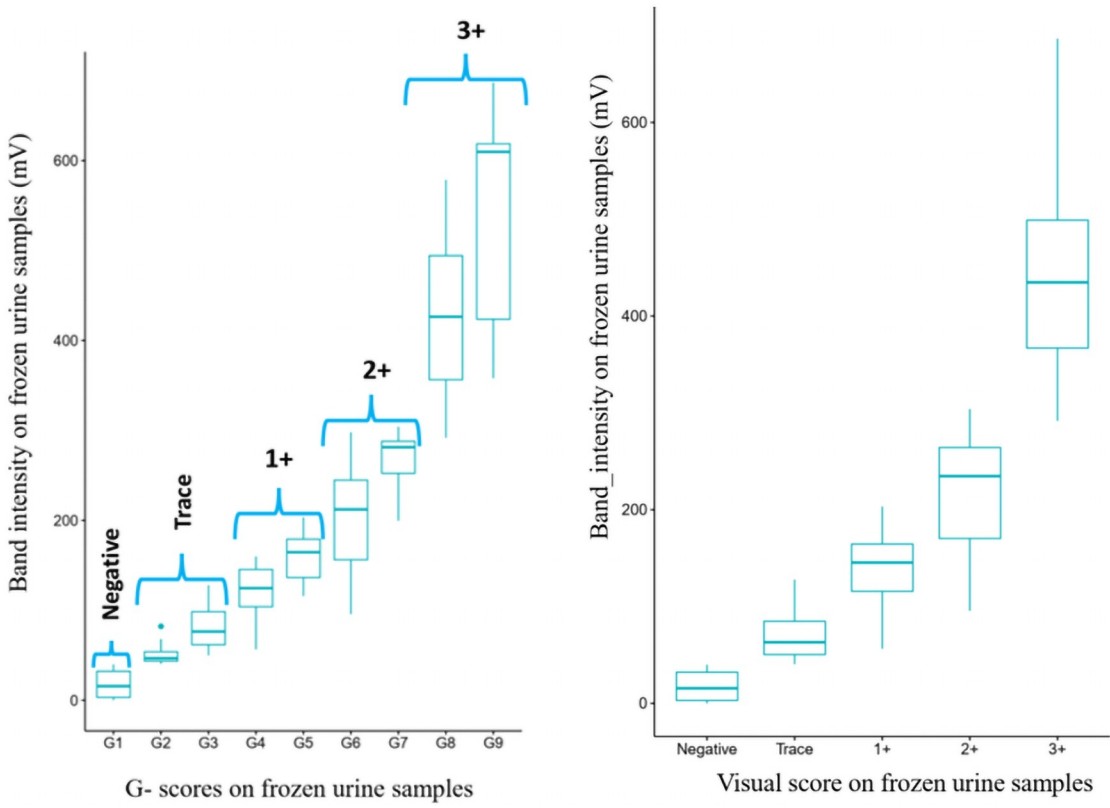

**Fig 6. Correlation between infection intensity from the POC-CCA tests using the ESEquant reader and the visual scoring system (Fig 6A: G-scores; Fig 6B: Corresponding visual score).**

between our result and previous ones could be linked to the behaviour of children in different epidemiological settings. In Makenene for instance, children of 5 to 9 years follow their elders in different risky activities such as swimming, washing clothes and fetching water. All SAC may therefore have the same level of exposure to infected snails and cercariae. Although no difference was observed between girls and boys for the *S. mansoni* prevalence, a significant association was reported between infection intensity and sex. Boys appear to have higher *S. mansoni* infection intensities than girls. These results are in agreement with those obtained in Yemen and Cameroon [54,55]. They could be explained, in part, by the fact that during the field surveys, boys were observed to be swimming regularly in the river compared to girls. Regular participation in such activities exposes them to infected snails and water more frequently and consequently, may increase the intensity of schistosome infections in boys compared to girls.

The ESEquant LR3 reader used in the present study appeared as a reliable tool for efficient quantification of CCA in urine samples. The strong positive correlation observed between the eggs count and the amount of CCA indicate that quantifying CCA on POC-CCA strip can be used to estimate the infection intensity of *S. mansoni*. The positive correlation obtained in the present study is in line with previous data [18,23,44,61]. Quantifying CCA in urine samples could be used to estimate the infection intensity and subsequently, the worm burden in infected individuals. Although visual scoring of POC-CCA strip using G-scores has been used for semi-quantification of CCA, some challenges regarding inter-operator variability in the interpretation of results have been raised [26]. In the present study, the use of ESEquant LR3

reader overcame these difficulties because it assigned numerical values to all positive POC-CCA including results from strips interpreted as "trace". In addition, the strong positive correlation (Spearman rho = 0.89) observed by quantifying the amount of CCA on POC-CCA strips using the ESEquant LR3 reader and the visual scoring suggests that the visual system of scoring could be substituted by the POC-CCA reader depending on what the data will be used for. The numerical values generated on POC-CCA strips interpreted as "trace" (visual scoring system) highlights the ability of the lateral flow strip reader to objectively quantify the CCA in urine samples of children with low intensity infections. The use of an electronic lateral flow strip reader may enable reliable and reproducible quantification of CCA on POC-CCA strips. This avoids the variations that are often observed in the visual interpretation of POC-CCA results. The POC-CCA reader has the advantage of objectively and numerically quantifying CCA in biological samples. As the amount of released CCA is directly related to the number of adult living worm producing this antigen, the variation in the amount of CCA between individuals should reflect the difference in worm burdens. In addition to its simplicity and ease of use, the lateral flow strip reader is less laborious and requires less time for samples' analyses compared to the KK assay. With this reader, the hands on time to prepare and read two POC-CCA strips can be estimated to about 3 mins. However, the lack of electricity in many schistosomiasis endemic areas and additional investment to purchase the reader that costs about 4233 euros are required compared to the visual scoring system.

Remarkably, 100 urine samples can be processed and read with POC-CCA reader by two individuals in less than 4 hours. Without any requirement of technician with specialized skills, the prevalence and the intensity of *S. mansoni* infections will therefore be obtained in a very short period. Three well trained technicians and more time will be required to process the same number of stool samples by the KK test. The time needed to prepare, read and count eggs on one KK slide is about 10 mins (1 min to prepare a slide and 2–16 mins to go through the slide under microscope). It would take at least 6 hours for three well trained technicians to process and read 100 KK slides. In addition to the time spent to prepare and read the KK slides, the slides will be left for one or more hours at ambient temperature as preconized by the WHO for the clarification of the slides before they could be visualized under microscope. In consequence, more technicians and more days in the field will be required if the KK test is used in comparison to POC-CCA. For both POC-CCA reader and the KK test, the electricity is needed. In view of the requirement for skilled technicians for the KK test and the number of days required for each field survey, the survey costs to determine the prevalence and the intensity of *S. mansoni* infections will often be higher when using the KK, particularly if another KK slide is examined even on the same stool sample. The CCA is more sensitive and directly measures the intensity of infection rather the egg count which is a proxy for the former. Also, the sample collection is less offensive and hazardous for both participants and staff. Although the KK test has been widely used for the evaluation of infection intensity, the numerical quantification of CCA using lateral flow reader constitutes a big step forward for reliable estimation of the worm burden.

The POC-CCA appears therefore as a reliable tool for large scale epidemiological surveys targeting both the prevalence and the intensity of *S. mansoni* infections. In epidemiological settings characterized by low disease prevalence and low infection intensity as revealed by KK, the use of POC-CCA reader could provide reliable data that will help to boost schistosomiasis control with the overarching target of achieving the WHO goal of controlling the disease by 2025. Surprisingly, the POC-CCA test revealed high levels of CCA indicating high worm burden in some children who were negative on the KK test. These results could be explained by the fact that all schistosomes excrete CCA while there is no egg excretion by immature worms and single sex infections [44] as well as very variable rates of egg excretion in faeces [46]. The

real epidemiological value of high levels of CCA in urine of children with no detected *S. mansoni* eggs in their stools remains to be determined by performing longitudinal studies where these children could be followed up.

Although urine samples stored at +4˚C for one week or at -20˚C for one year have been recommended for POC-CCA test according to the manufacturer [62], results of the present study have shown that the mean value of CCA decrease when urine samples have been stored at -20˚C for one year. This had significant effect on the positive rate as almost a half of samples that were recorded as trace infections in fresh samples were negative in frozen samples. Nevertheless, it is important to point out that results of the present study revealed good correlations between data from POC-CCA reader and visual scoring system on fresh and frozen urine samples. The good correlations reported between data generated by the reader (Pearson coefficient = 0.85) and those from visual scoring system (Pearson coefficient = 0.84) on fresh and frozen samples demonstrates the capacity of POC-CCA to reproduce similar results on samples stored at -20˚ C for one year. In addition, when comparisons were performed between results generated by the POC-CCA reader and visual scoring system, good correlations were obtained for fresh (Spearman coefficient = 0.89) and stored (Spearman coefficient = 0.97) urine samples. In the present study, POC-CCA strips from fresh and stored urine samples were scored not only by different individuals but also at different time points. In such context, the probability of having some variations in the reading and scoring of POC-CCA strips from fresh and stored urine samples is high. The differences observed between results from the visual scoring system on fresh and stored urine samples could be also due to inter-variability in the reading of G scores by different individuals. The decrease in the amount of CCA after storage was probably expected due to antigen that could be degradated.

The fact that 89% of negative frozen samples were scored as "trace" (low quantity of CCA) on fresh urine samples raises the following questions: i) what could be the real epidemiological value of results interpreted as "trace"; ii) what positivity threshold must be considered for POC-CCA reader. Although some of these negative frozen samples, especially those with "trace" scores on fresh samples might not have had schistosomiasis, it is most likely that these changes in results are due to degradation of CCA in storage. This could occur in remote areas of most endemic developing countries where light shortage is common and the freezers can experience power outages leading to several freeze-thaw cycles of samples. Results of the 6 negative frozen samples previously scored on fresh samples as 1+, 2+ and 3+ are more difficult to interpret. Explanations include variation in sample treatment before storage, sample mix-ups and interference from other factors in the urine.

## Conclusion

This study confirmed the low sensitivity of the KK compared to POC-CCA in the estimation of the prevalence and intensity of *S. mansoni* infections. POC-CCA reader can be used in the field for numerically and reproducible quantification of CCA in urine samples. The numerical values generated by this reader make it possible to avoid the use of subjective visual interpretation of POC-CCA results. Results of this study revealed children with high level of CCA in their urine or high intensity of *S. mansoni* infections, but without *S. mansoni* eggs in their stools. The good correlation observed between results obtained on fresh and frozen urine samples confirm that POC-CCA test can be used on sample stored for one year at -20˚ C. Additional investigations are required to understand the real epidemiological value of high levels of CCA in children with no detectable *S. mansoni* eggs in their stools.

## Acknowledgments

We are grateful for the support and participation of the study populations included in this Study.

## Author Contributions

**Conceptualization:** Estelle Mezajou Mewamba, Harry Noyes, Hilaire Marcaire Womeni, Flobert Njiokou, Gustave Simo.

**Data curation:** Estelle Mezajou Mewamba, Arnol Auvaker Zebaze Tiofack, Cyrille Nguemnang Kamdem, Romuald Isaka Kamwa Ngassam, Mureille Carole Tchami Mbagnia, Harry Noyes, Gustave Simo.

**Formal analysis:** Cyrille Nguemnang Kamdem, Mureille Carole Tchami Mbagnia, Oscar Nyangiri, Harry Noyes, Hilaire Marcaire Womeni, Gustave Simo.

**Funding acquisition:** Gustave Simo.

**Methodology:** Estelle Mezajou Mewamba, Arnol Auvaker Zebaze Tiofack, Cyrille Nguemnang Kamdem, Romuald Isaka Kamwa Ngassam, Mureille Carole Tchami Mbagnia, Oscar Nyangiri, Harry Noyes.

**Project administration:** Gustave Simo.

**Supervision:** Hilaire Marcaire Womeni, Flobert Njiokou.

**Writing – original draft:** Arnol Auvaker Zebaze Tiofack, Romuald Isaka Kamwa Ngassam, Flobert Njiokou, Gustave Simo.

**Writing – review & editing:** Oscar Nyangiri, Harry Noyes, Gustave Simo.

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
