## [Decision Letter · Decision Letter 0]

3 Apr 2021

Dear Dr Simo,

Thank you very much for submitting your manuscript "Field assessment in Cameroon of a reader of POC-CCA lateral flow strips for the quantification of Schistosoma mansoni Circulating Cathodic Antigen in urine" for consideration at PLOS Neglected Tropical Diseases. As with all papers reviewed by the journal, your manuscript was reviewed by members of the editorial board and by several independent reviewers. The reviewers appreciated the attention to an important topic. Based on the reviews, we are likely to accept this manuscript for publication, providing that you modify the manuscript according to the review recommendations. 

Sincerely,

Jessica K Fairley, MD, MPH

Associate Editor

Michael Hsieh

Deputy Editor

Reviewer's Responses to Questions

**Key Review Criteria Required for Acceptance?**

**Methods**

-Are the objectives of the study clearly articulated with a clear testable hypothesis stated?

-Is the study design appropriate to address the stated objectives?

-Is the population clearly described and appropriate for the hypothesis being tested?

-Is the sample size sufficient to ensure adequate power to address the hypothesis being tested?

-Were correct statistical analysis used to support conclusions?

-Are there concerns about ethical or regulatory requirements being met?

Reviewer #1: Introduction

L90: Why (sub)?

L100: (SAC)

L102: After introducing the abbreviation PZQ please continue to use it ot not introduce it at all

L104: Rather than 'the real' I suggest 'accurate'

L106: You are not just monitoring prevalence but also 'intensity'. I suggest you use the term progress towards control.

L110: I don’t agree it is especially low-cost as training and retaining capable microscopists is expensive. You have not described the cost of POC-CCA or the LR3 reader for comparison.

L112: How important are these low intensity infections in terms of control defined as <1% moderate and high intensity infections (MHI). If an infection in undetectable by KK hasn’t ‘control’ has already been achieved? Are you suggesting elimination is the gaol?

L120: Stool 'or urine'.

L131: As stated previously there appears to be 4 objectives, please deal with them separately in this last sentence.

L135: The roman numbering is unnecessary in these headings, please delete.

Methods

L144: What about younger children?

L146: In the study

L150: WHO recommendations

L165: Has there been any Schistosomiasis intervention in this area?

L166: Readers will be interested in the range of prevalence and intensity found previously

L169-172: Please provide information on how these schools and participants selected (convenience or random sampling)?

L190 and 194: Stick with the abbreviation KK

L222: L 220-223 A figure illustrating the LR3 reader would be appreciated.

L234 and L235: Grammar check.

L271 and 276: What do you mean by age group. Table 3 is not showing age groups but ages.

L279-280: Of the 71 by KK.

L285: CCA

L304: Prevalence and intensity of S. mansoni by.

L308 Table 3: Consider presenting this as age groups 5<10 years and 10<15 years the stat might be more interesting and a detailed describtion of epidemiology was not one of the main objectives

Table 2 and 3 Title: ‘Prevalence and intensity of…..

Since examination of prevalence and intensity by age and by sex was not a defined objective perhaps the results should be more simply described by age group in Table 3 as it detracts for the other objectives of evaluating the LR3 reader in comparison with KK

L312: No need of 'the abundance'

L326: CCA

L331: This classification is expressed differently in methods: light (<100) moderate (100-400) and heavy (>400). Please concur with the WHO classification.

Lines 344 to 362 should be greatly simplified as table 4 illustrates the findings.

Reviewer #2: Yes

Reviewer #3: It appears that there are prior studies that have used either KK or CCA to estimate prevalence in these communities. Can the authors state those prior estimates to give the reader and initial understanding about the appropriateness of sample size?

**Results**

-Does the analysis presented match the analysis plan?

-Are the results clearly and completely presented?

-Are the figures (Tables, Images) of sufficient quality for clarity?

Reviewer #1: L385: Please describe the study limitations 

L388: Prevalence 'and intensity' or perhaps 'disease burden'

L392-394: Could you please rephrase this sentence is not clear.

L403: It would be nice to give possible reasons for these risky activities. What is the situation like in Makenene?

L410: What do you mean by 'younger' pre-SAC or the age group 5<10?

L412: SAC

L415: No need for 'or worm burden'

L416: Rather than 'previous authors it is easier for the reader if you say 'where'

L417: The discussion on age and sex could be more concise as the 'participation in risky behavior' is repetitious.

L445: KK

L447: You have made no mention of costs of the lR3 reader and until now no mention that it requires electricity which may be unavailable in many POC settings.

L457-458: I know the cellophane has to be soaked for 24hrs to be used then slides can be read as long as the faecal smear is spread evenly on the slide. Please clarify.

L464: Does a second slide need to be taken from a second stool sample. Isn't another part of the same stool sample sufficient?

L465: Do you mean intensity of infection? Doesn’t worm burden imply both prevalence and intensity combined?

L474: Is the WHO goal for SCH to eliminate or to control? If it is to control then low levels of prevalence and intensity have already met that goal.

L480: Why use schistosome here rather than S. mansoni?

L484: Delete 'seems to'

L485: The overly complicated presentation in the results camouflaged this important point

L492: Again I am sure you can find a way of expressing this in a more concise clear manner in both results and here in discussion.

L503: Again simplify the results and this discussion of results as it is overly complicated

Reviewer #2: Yes

Reviewer #3: The analysis is clearly presented and aligns well with the analysis plan stated. A valuable addition to the paper would be a visual comparison of the CCA cassettes that were classified as 0, “trace”, 1+, 2+ or 3+. Interpretation of the results would be augmented by an understanding if mass drug administration of praziquantal has been ongoing in this sample prior to the study.

**Conclusions**

-Are the conclusions supported by the data presented?

-Are the limitations of analysis clearly described?

-Do the authors discuss how these data can be helpful to advance our understanding of the topic under study?

-Is public health relevance addressed?

Reviewer #1: Please describe study limitations

Conclusion

L 514: The whole paper need to be reorganized along the 4 (possible 5 if you include epidemiological description by age and sex) objectives and then the conclusion should follow this same sequence of organization and simplified interpretation if SCH control is defined as <1% MHI.

Reviewer #2: Yes

Reviewer #3: This paper does an excellent job of presenting final conclusions that are aligned with the data presented. CCA has enormous potential to improve diagnostics if it can determine intensity of infection as presented in this paper.

On line 417 of the Discussion it states "In the present study, boys were found to swim regularly in the river." It is not clear how this observation was made. How were the data collected? If by questionnaire please add a table to describe these behaviors. If anecdotal please state it as such rather than as part of this study.

In Reference to the discussion of keeping the urine samples frozen at -20C on Line 485, can the authors discuss the potential for antigen degradation in the frozen urine samples if the freezers experienced power outages or other technical issues? It will be helpful to know if the samples analyzed when through freeze-thaw cycles.

**Editorial and Data Presentation Modifications?**

Reviewer #1: Abstract L39: There seems to be at least 4 objectives in this study 1) compare prevalence by CCA with KK, 2) compare visual appraisal of CCA with the LR3 reader 2) compare fresh and frozen results and 4) compare intensity of infection by LR3 reader with KK. Please clarify this in the abstract and make adjustments to the background, methods, methods, results and conclusion.

L 44: Describe the LR3 reader as a means of quantification of the degree of positively

L48: Visual scoring of intensity by KK (epg) or (ep10ml?) versus visual scoring of POC-CCA. please clarify

L51: pleased define frozen (<-20C)

L58: what about the last objective comparing LR3 reader and KK for intensity?

Author Summary

L65: Spell POC-CCA in full for the first time of use

L66: Now you can use CCA

L68-70 there were at least 4 objectives to this study. Best to clarify that from the start and deal with each separately as you proceed

L72: SAC defined by 5-15 years?

L77: Better to define frozen at -20C earlier in this sentence and in the full abstract.

Discussion

L385: Please describe the study limitations 

L388: Prevalence 'and intensity' or perhaps 'disease burden'

L392-394: Could you please rephrase this sentence is not clear.

L403: It would be nice to give possible reasons for these risky activities. What is the situation like in Makenene?

L410: What do you mean by 'younger' pre-SAC or the age group 5<10?

L412: SAC

L415: No need for 'or worm burden'

L416: Rather than 'previous authors it is easier for the reader if you say 'where'

L417: The discussion on age and sex could be more concise as the 'participation in risky behavior' is repetitious.

L445: KK

L447: You have made no mention of costs of the lR3 reader and until now no mention that it requires electricity which may be unavailable in many POC settings.

L457-458: I know the cellophane has to be soaked for 24hrs to be used then slides can be read as long as the faecal smear is spread evenly on the slide. Please clarify.

L464: Does a second slide need to be taken from a second stool sample. Isn't another part of the same stool sample sufficient?

L465: Do you mean intensity of infection? Doesn’t worm burden imply both prevalence and intensity combined?

L474: Is the WHO goal for SCH to eliminate or to control? If it is to control then low levels of prevalence and intensity have already met that goal.

L480: Why use schistosome here rather than S. mansoni?

L484: Delete 'seems to'

L485: The overly complicated presentation in the results camouflaged this important point

L492: Again I am sure you can find a way of expressing this in a more concise clear manner in both results and here in discussion.

L503: Again simplify the results and this discussion of results as it is overly complicated

Reviewer #2: Specific comment and revision

Page 13, line 143-144: author mentioned that parents or guardians signing consent on behalf of the children and also mention some set of children signing an assent form, this is a contradicting statement.

Page 14, line 168-172: author did not state how the sample size was calculated and determine.

Page 15, line 188-202: KK examination was carried out on fresh stool samples just once, subsequence examination should have been employed there is disparity in the number of eggs per gram in the stool samples.

Page 14, line 169- 172: numbers of samples collected at each school were not stated.

Page 16, line 228: The specific type of drawer where POC-CCA cassette was removed wasn’t well stated, unlike line 222, which state each cassette was introduced into the drawer of the ESEQuant

Page 16, line 229:” G-scores as described by”…not well stated.

Reviewer #3: No minor modifications needed.

**Summary and General Comments**

Reviewer #1: Overall, I wonder what the paper is trying to convey. Are you proposing the use of the LR3 reader as a substitute for KK (microscopy)? If so, please provide more details about costs and the need for electricity. Why spend so much attention on fresh v frozen samples? If the LR3 reader were to be used as a substitute for KK it would be used as a POC tool? The degradation of accuracy with freeze/thaw is to be expected but I don’t think it warrants the extensive point by point description in the results and discussion which need to be made more concise. Finally, the WHO goal of control as public health significance (<1% MHI) and the role CCA and the LR3 reader might play in helping countries reach that goal in resource poor settings would be extremely helpful.

Reviewer #2: This study has confirmed the low sensitivity of KK and the high capacity of POC-CCA test to provide reliable diagnostic data on schistosomiasis prevalence, particularly in low endemic settings. Also, it has shown the lateral flow reader enabled accurate quantification of CCA under field conditions on fresh and frozen urine samples with less time and effort than KK. Also, the author has been able to confirm that POC-CCA test can be used on samples stored for one year at -200 C.

Reviewer #3: The authors have presented an important piece of work to advance the use of CCA from a qualitative binary diagnostic to a quantitative diagnostic of intensity of infection. This is an essential tool to eliminate Schistosomiasis from these communities and as MDA are rolled out across endemic countries these intensity measurements will become more important. I applaud the authors for their clearly written and presented results. Prior understanding of MDA exposure in this population will help readers to interpret the age-stratified intensities presented in this work.

PLOS authors have the option to publish the peer review history of their article (what does this mean?). If published, this will include your full peer review and any attached files.

Reviewer #1: Yes: Dr Mary H. Hodges

Reviewer #2: No

Reviewer #3: No

Figure Files:

Data Requirements:

Reproducibility:

References

---

## [Editor Report · Decision Letter 1]

16 Jun 2021

Dear Dr Simo,

We are pleased to inform you that your manuscript 'Field assessment in Cameroon of a reader of POC-CCA lateral flow strips for the quantification of Schistosoma mansoni Circulating Cathodic Antigen in urine' has been provisionally accepted for publication in PLOS Neglected Tropical Diseases.

Best regards,

Jessica K Fairley, MD, MPH

Associate Editor

Michael Hsieh

Deputy Editor

---

## [Editor Report · Acceptance letter]

30 Jun 2021

Dear Dr Simo,

We are delighted to inform you that your manuscript, "Field assessment in Cameroon of a reader of POC-CCA lateral flow strips for the quantification of Schistosoma mansoni Circulating Cathodic Antigen in urine," has been formally accepted for publication in PLOS Neglected Tropical Diseases.

Best regards,

Shaden Kamhawi

co-Editor-in-Chief

Paul Brindley

co-Editor-in-Chief
